# Healthcare Access Among Individuals Who Practice Chemsex in Brazil: A Scoping Review Protocol

**DOI:** 10.3390/nursrep15100353

**Published:** 2025-09-27

**Authors:** Isadora Silva de Carvalho, Lariane Angel Cepas, Álvaro Francisco Lopes de Sousa, Talita Morais Fernandes, Talia Gomes Luz, Jean Carlos Soares da Silva, Augusto da Silva Marques, Caíque Jordan Nunes Ribeiro, Shirley Veronica Melo Almeida Lima, Anderson Reis de Sousa, Carlos Arterio Sorgi, Ricardo Nakamura, Ana Paula Morais Fernandes

**Affiliations:** 1Department of General and Specialized Nursing, Ribeirão Preto School of Nursing, University of São Paulo, Ribeirão Preto 14040-902, SP, Brazil; isadoracarvalho@usp.br (I.S.d.C.); taliagomesluz@usp.br (T.G.L.); jeancarloseerp@usp.br (J.C.S.d.S.); silva75@usp.br (A.d.S.M.); anapaula@eerp.usp.br (A.P.M.F.); 2Institute of Teaching and Research, Hospital Sírio-Libanês, São Paulo 01308-050, SP, Brazil; 3Public Health Research Centre, Comprehensive Health Research Center (CHRC), NOVA University Lisbon, 1099-085 Lisbon, Portugal; 4Ribeirão Preto Faculty of Medicine, Universidade de São Paulo, Ribeirão Preto 05508-900, SP, Brazil; talitafernandes@usp.br; 5Graduate Program of Nursing, Federal University of Sergipe, São Cristóvão 49107-230, SE, Brazil; caiquejordan@academico.ufs.br (C.J.N.R.); shirleymelo.lima@gmail.com (S.V.M.A.L.); 6School of Nursing, Federal University of Bahia, Salvador 40170-110, BA, Brazil; anderson.sousa@ufba.br; 7Department of Chemistry, Ribeirão Preto Faculty of Philosophy, Science, and Letters, University of São Paulo, Ribeirão Preto 14040-900, SP, Brazil; carlos.sorgi@usp.br; 8Department of Computer Engineering and Digital Systems, Polytechnic School, University of São Paulo, São Paulo 05508-010, SP, Brazil; ricardonakamura@usp.br

**Keywords:** chemsex, sexualized drug use, healthcare access, harm reduction, Brazil, Unified Health System (SUS), nursing

## Abstract

**Background:** Chemsex, the intentional use of psychoactive substances to enhance sexual experiences, is an emerging public health issue in Brazil, associated with increased risks of sexually transmitted infections and complex psychosocial vulnerabilities. Despite the universal coverage provided by the Unified Health System (SUS), individuals who practice chemsex often encounter barriers to healthcare, including stigma, discrimination, and a lack of specialized services. To date, no comprehensive reviews appear to synthesize evidence on how this population accesses healthcare in the Brazilian context; existing knowledge remains fragmented across individual studies. **Objectives:** The aim is to map and synthesize the available evidence regarding access to health services among people who engage in chemsex in Brazil, identifying health needs, professional demands, barriers, and facilitators. **Methods:** The protocol follows the Joanna Briggs Institute methodology for scoping reviews and PRISMA-ScR guidelines. A systematic search will be conducted in MEDLINE (PubMed), Embase, Scopus, SciELO, and LILACS for studies published between 2014 and 2024 in Portuguese, English, or Spanish. Data will be summarized using descriptive and narrative synthesis, presented in tables and thematic categories. Studies will be included if they address chemsex or sexualized drug use in Brazil and report on healthcare access, regardless of gender identity, sexual orientation, or drug type. Studies that do not address chemsex, focus on drug use outside a sexual context, or are unrelated to Brazil will be excluded. **Expected results:** The review is expected to identify key barriers and facilitators to healthcare access, highlight knowledge gaps for underrepresented groups, and support recommendations for research, policy, and practice to improve care for people engaging in chemsex in Brazil. By focusing on an underexplored intersection of drug use, sexuality, and healthcare access in Latin America, this study aims to provide an innovative contribution to public health literature.

## 1. Introduction

Chemsex, a term derived from the combination of the words “chemical” and “sex,” has garnered increasing attention as a global public health concern [1], reflecting the complex interactions between sexual and social dynamics and the use of psychoactive substances among diverse populations [2]. Historically, chemsex has emerged in contexts marked by marginalization, discrimination, and limited social support, being particularly prevalent among sexual and gender minorities who experience institutional and social exclusion [3,4].

The concept of chemsex was introduced by David Stuart in the 1990s to describe the phenomenon among gay men in London [5]. Within this sociocultural landscape, the intentional use of specific drugs aimed to facilitate, enhance, or prolong sexual experiences, often during group encounters or private parties [5,6]. For many individuals, chemsex represented not only a pursuit of pleasure but also a response to stigma, social isolation, and the need for belonging in a period when traditional spaces were inaccessible or hostile to sexual diversity.

Therefore, chemsex emerges as a socially embedded phenomenon, closely linked to structural conditions of exclusion, vulnerability, and the search for community, pleasure, and acceptance [6,7]. Its features and expressions have been shaped by the historical and sociocultural contexts of groups that, in the face of discrimination, developed new forms of sociability and mutual support mediated by substance use [5,7].

However, the definition of chemsex remains multifaceted and subject to ongoing debate in the scientific literature [2,8,9], influenced by social, historical, and economic factors, as well as by the availability of substances in different regions and time periods. Initially, the term referred to the use of a limited set of drugs—primarily methamphetamine, GHB/GBL, mephedrone, and ketamine—predominantly observed in urban European settings, especially among gay men in cities such as London [2,8,9]. This restricted definition reflects both epidemiological patterns and cultural and economic factors that influence which substances are accessible, accepted, or criminalized in specific contexts [10].

Recent literature has proposed expanding the definition to include any substance capable of altering consciousness or sexual performance—such as alcohol, cocaine, cannabis, poppers, ecstasy, and even medications for erectile dysfunction [8,11]—provided their use is intentional and linked to sexual contexts. This broader perspective underscores that chemsex is a dynamic phenomenon, shaped by time, space, economic conditions, and the cultural and social transformations of the groups involved.

It is also important to distinguish chemsex from what is referred to as “sexualized drug use” (SDU). While chemsex, in its original definition, is associated with the use of specific substances in sexual contexts, generally during group or prearranged encounters, SDU refers more broadly to the use of any drug immediately before or during sexual activity, encompassing a wide range of profiles, motivations, and practices [1,4]. Although not all instances of SDU constitute chemsex in the strict sense, both phenomena share vulnerabilities, risks, and challenges within the field of public health and in relation to access to care.

The motivations for engaging in chemsex are diverse and complex, including the pursuit of enhanced pleasure, disinhibition, social connection, reduced anxiety, prolonged sexual performance, and coping strategies for psychological distress, isolation, or stigma [12,13]. Nevertheless, chemsex is strongly associated with a range of biological and psychosocial risks, including higher prevalence of sexually transmitted infections, episodes of sexual violence, substance misuse and dependence, mental health disorders, suicide attempts, and increased social vulnerability [14,15].

In Brazil, the analysis of chemsex requires careful consideration of sociocultural factors, legislation, and the organization of the public health system [8,16]. The 1988 Federal Constitution guarantees universal and free access to healthcare, operationalized through the Unified Health System (Sistema Único de Saúde—SUS), which is founded on the principles of universality, equity, and comprehensiveness. In practice, however, the implementation of these principles and of policies related to harm reduction, sexual and reproductive health, and the care of vulnerable populations varies considerably across states and municipalities, and over time [17].

Evidence regarding access to healthcare services among people who engage in chemsex in Brazil remains limited [2]. Available studies suggest that access may be hindered by stigma, institutional discrimination, lack of provider training, and the scarcity of welcoming or specialized services [18,19]. Furthermore, Brazilian scientific production on chemsex has been concentrated mainly on men who have sex with men [2,11], with notable gaps concerning other groups and an overemphasis on biomedical aspects. Psychosocial, cultural, historical, and structural dimensions of the phenomenon are often underexplored, and there are few investigations into care-seeking trajectories, service engagement and satisfaction, or strategies to overcome healthcare barriers.

By contrast, the international literature has made greater progress in describing consumption patterns, motivations, risks, and outcomes related to chemsex, providing evidence that can inform public policies, clinical practices, and harm reduction strategies [20,21,22].

## 2. Materials and Methods

This scoping review study is based on the principles outlined in the JBI Reviewer’s Manual: (1) defining and aligning the review objectives and questions, (2) developing and aligning the inclusion criteria with the objectives and questions, (3) developing a search strategy, (4) searching for evidence, (5) selecting the evidence, (6) extracting the evidence, (7) analyzing the evidence, (8) presenting the results, and (9) summarizing the evidence in relation to the review purpose, drawing conclusions, and considering the implications of the findings [23].

The Population, Concept, and Context (PCC) framework was used to define the review objectives, research questions, and search strategy, with P (Population) = individuals who engage in chemsex, C (Concept) = healthcare access, and C (Context) = Brazil.

### 2.1. Participants

Studies addressing all populations engaging in chemsex or any form of SDU in Brazil will be included, regardless of gender identity, sexual orientation, or social group. Chemsex is often associated with men who have sex with men (MSM) and a restricted set of substances (such as methamphetamine, GHB/GBL, mephedrone, and ketamine) [12,24,25,26], but this review aims to include any individual who associates the use of psychoactive substances with sexual activity, irrespective of the substance used. The literature presents challenges in characterizing this population, with inconsistent definitions and limitations that exclude groups such as cisgender and transgender women, non-binary individuals, and heterosexual persons who also engage in SDU. Therefore, this review will adopt broad inclusion criteria to ensure the incorporation of diverse experiences within the phenomenon of chemsex.

To address heterogeneity and avoid overgeneralization, data on populations and substance use will be analyzed according to two predefined categories during the synthesis of results. Although chemsex does not have a universally defined set of substances and can involve a wide variety of psychoactive drugs, for the purposes of this review and to facilitate comparison and discussion with existing literature, a stratification approach will be applied. Chemsex-narrow will include the use of stimulants or specific substances commonly associated with chemsex (e.g., methamphetamine, GHB/GBL, mephedrone, ketamine), while SDU-broad will encompass other substances used in sexual contexts, including alcohol, cannabis, poppers, and similar substances. This stratified approach will ensure that findings related to chemsex-narrow are presented separately from SDU-broad, allowing for accurate interpretation of results, minimizing the risk of overgeneralization, and providing a clear framework for discussion in relation to other studies.

Studies that do not directly address chemsex, only mention drug use unrelated to sexual contexts, or focus solely on populations outside the Brazilian context will be excluded.

### 2.2. Concept

The central concept of this review is access to healthcare for people who engage in chemsex in Brazil. To define access, we will use the model proposed by Botelho and França Júnior [27], which structures access to health services across different dimensions:Availability and supply—the existence of adequate and accessible services for this population.Accessibility—geographical, economic, and structural barriers that affect access.Accommodation—the capacity of services to meet the needs of this population.Acceptability—the perceptions and experiences of people who engage in chemsex regarding the care received.Effective use—the effectiveness of services in promoting the health of this population.

Access to healthcare will be analyzed considering both the structure of the SUS and the social and cultural factors that influence healthcare seeking and retention. Studies exclusively addressing drug use unrelated to healthcare seeking, or those limited to a biomedical perspective without considering structural and social aspects of access, will be excluded.

## 3. Context

In Brazil, health is recognized as a fundamental right, as established by the 1988 Federal Constitution [28], and is guaranteed through the SUS, which is universal, free of charge, and nationally implemented. Brazilian legislation and SUS guidelines emphasize the principles of universality, equity, and comprehensive care, aiming to ensure that all individuals—regardless of sexual orientation or gender identity—have access to appropriate and high-quality healthcare. As one of the largest public health systems in the world, the SUS has been central to disease prevention, health promotion, and epidemic response. However, the availability and implementation of services, including those related to HIV/AIDS, mental health, and substance use, vary considerably across states and municipalities and have changed over time. In the context of chemsex, it is hypothesized that the SUS could play an important role in harm reduction strategies, sexual health services, and psychosocial support, but the extent and effectiveness of this role remain to be mapped and critically examined.

The PCC acronym (Population, Concept, Context) guided the formulation of the primary research question: What is known about healthcare access among individuals who practice chemsex in Brazil? In addition to the main question, secondary questions were developed to deepen the analysis of the findings and to structure data extraction, allowing for a more comprehensive understanding of different dimensions of the phenomenon: 1. What are the needs of individuals who practice chemsex and of healthcare professionals in relation to accessing healthcare services in Brazil? 2. Which existing barriers and facilitators affect the access of individuals who practice chemsex in Brazil to healthcare services? 3. What services and/or specific resources are available within the Brazilian healthcare system to provide care for individuals who practice chemsex?

These questions will guide the search for articles in the MEDLINE/PubMed, Embase, Scopus, SciELO, LILACS, and PsycINFO databases. The initial search strategy was developed in PubMed using the PCC framework and was based on relevant keywords combined with the Boolean operators “AND” and “OR” (see Appendix A). This initial strategy was subsequently adapted for each selected database, taking into account the specific indexing terms, search language, and system characteristics of each platform. The final strategies were reviewed and refined with the support of a specialized librarian to ensure both accuracy and comprehensiveness.

The inclusion criteria to be applied will be: primary studies published between 2014 and July 2025, in Portuguese, English, or Spanish, and indexed in the selected databases. Exclusion criteria will include: study protocols, systematic reviews, studies whose titles and abstracts do not address the research question, studies that do not include the perspectives of individuals who engage in chemsex or of healthcare professionals providing care to this population, studies not conducted within Brazil’s public healthcare system, studies not available in full text, and duplicates across databases. Experience reports, opinion articles, editorials, grey literature, and other publications that do not present empirical data will also be excluded.

All database searches will be documented, detailing the database consulted, search date, search strategy (combination of keywords), applied limits, and the number of articles retrieved.

In the next step, all references will be exported to Covidence©. The review process will consist of two screening stages. The first stage will involve screening titles and abstracts to assess eligibility. The second stage will consist of full-text reading of studies selected in the first stage. In both stages, two or more independent reviewers will conduct the selection process. Disagreements will be resolved through discussion or, if necessary, by involving a third reviewer until consensus is reached.

The reporting of the review will adhere to the Preferred Reporting Items for Systematic Reviews and Meta-Analyses Extension for Scoping Reviews (PRISMA-ScR) [29]. A PRISMA flow diagram will be provided to illustrate the selection process.

Key information from each selected study will be synthesized using a data charting form specifically developed for this review, based on the JBI framework [23], to support descriptive and narrative analyses. Data charting will be carried out by one reviewer and verified by a second reviewer to ensure accuracy and consistency.

The data charting form will include the following fields: year of publication; language of publication; study location; study type; population studied; and main themes.

In accordance with the JBI approach, a consultation phase will be conducted to enhance the comprehensiveness of the review through the identification of additional sources of data. This process will include citation analysis of the studies previously selected from the database searches. In this phase, the Scopus database will be used, as it provides advanced tools for citation tracking. All titles and abstracts of works citing the previously included studies will be identified. Subsequently, the titles and abstracts of these citing articles will be screened using the same pre-established inclusion and exclusion criteria. This strategy will allow the identification of relevant studies that may not have been captured during the initial database searches. Levac, Colquhoun, and O’Brien emphasize that the consultation phase is essential in scoping reviews, as it enriches the study by incorporating literature that increases the representativeness and depth of the investigated corpus [30].

The PRISMA-ScR guidelines [29] will be followed to systematize the process of study inclusion in this review. A flow diagram outlining this process will be presented alongside the results.

## 4. Expected Results

To address the research questions previously defined, the results of this scoping review will be presented using a descriptive and narrative approach, in accordance with the JBI methodology. Quantitative data from the included studies—such as year of publication, geographic distribution (city, state, and region), methodological design, and characteristics of the studied population (individuals who engage in chemsex)—will be summarized using absolute and relative frequencies and displayed in tables and figures to highlight general patterns. In parallel, the main findings of the studies will be organized into thematic categories, developed both inductively and deductively based on the research questions. These themes will be used to explore barriers, facilitators, and contextual aspects related to chemsex in the Brazilian context, as reported by participants, healthcare professionals, health services, or the broader healthcare system.

## 5. Ethical Aspects

This study, being a scoping review, does not require approval from a Research Ethics Committee, in accordance with the guidelines established by Resolution No. 510/2016 of the Brazilian National Health Council (CNS).

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
