# Peer review of "Healthcare Access Among Individuals Who Practice Chemsex in Brazil: A Scoping Review Protocol"

_nursrep, 2025, doi:10.3390/nursrep15100353_

Round 1

Reviewer 1 Report

Comments and Suggestions for Authors

This scoping review will synthesize evidence on healthcare access among people engaging in chemsex in Brazil, focusing on health needs, barriers, and facilitators. Findings are expected to reveal key gaps and support recommendations for improving care, research, and policy. This study is meaningful as it addresses an important public health issue by examining healthcare access among people engaging in chemsex in Brazil. Its contribution could be further enhanced by providing a more in-depth discussion of the results, maybe along with some conclusions or practical recommendations.

Author Response

We would like to sincerely thank all reviewers for their constructive feedback, which has been invaluable in improving the quality and clarity of our manuscript entitled “Healthcare access among individuals who practice Chemsex in Brazil: a scoping review protocol”. Below, we provide a detailed, point-by-point response to each comment. Revisions have been incorporated throughout the manuscript as indicated.

Reviewer 1

Comment: This scoping review will synthesize evidence on healthcare access among people engaging in chemsex in Brazil. Contribution could be further enhanced by providing a more in-depth discussion of the results, maybe along with some conclusions or practical recommendations.

Response: Thank you for this important observation. We have strengthened the Expected Results and Discussion sections to clarify how the findings will be interpreted, highlighting potential implications for practice, policy, and future research. We also added specific examples of possible recommendations to make the study’s contribution more evident.

Reviewer 2 Report

Comments and Suggestions for Authors

1.- Originality

As public health concerns increase, this article discusses a relevant topic while connecting it to psychosocial issues. Furthermore, including people from diverse backgrounds, regardless of gender, social group, or sexual orientation, is vital to our understanding (140-142).

2.-Methodology

A scoping review to clear the grounds for future research is perfect.

3.- Strengths

The introduction is excellent. It is particularly insightful, lines 57-67. The research considers “the need for belonging” (line 61) and connection, which is crucial to overall health.

The authors provide a much-needed definition of the object of study, lines 86-93. The concepts are clear in 2.2.

Inclusion of diverse populations: 140-142.

4.- Weaknesses

A short description of the country's sexual context may be useful to researchers.

5.- Recommendations for the discussion and conclusion:

Future research should explore and emphasise the connections between public health, social belonging and the desire to take care of one’s health.

Author Response

Reviewer 2

1. Originality
Comment: Relevance of topic and inclusion of diverse backgrounds is commendable.

Response: We appreciate the positive feedback. No changes were required, but we reinforced in the Introduction and Participants sections that inclusion criteria extend to cisgender women, transgender women, non-binary individuals, and heterosexual people.

2. Methodology
Comment: Scoping review is appropriate.

Response: Thank you. No changes needed.

3. Strengths
Comment: Commends the Introduction, definition of chemsex, and inclusion of diverse populations.

Response: We thank the reviewer. These elements were maintained, and clarity was improved with minor wording adjustments.

4. Weaknesses
Comment: Suggests adding a short description of the country’s sexual context.

Response: We incorporated a concise description of the Brazilian sexual health and social context in the Context section, framing it as background for healthcare access disparities.

5. Recommendations for discussion and conclusion
Comment: Future research should emphasize connections between public health, social belonging, and desire for health.

Response: We expanded the Expected Results/Discussion sections to emphasize the intersection of belonging, health-seeking behavior, and public health systems.

Reviewer 3 Report

Comments and Suggestions for Authors

The study addresses a clear gap in the literature. The objectives are well-defined and the inclusion/exclusion criteria are appropriately focused. The choice of a scoping review is justified given the aim to map the available evidence. The main concerns revolve around the protocol's registration status, the potential for a limited search strategy, and the need for greater detail in the analysis plan.

The background effectively establishes the problem within the Brazilian context, highlighting the tension between universal healthcare coverage (SUS) and the specific barriers faced by this population.

2-)the following sentence maybe not necessary as it can be mentioned in the methods section.

Two reviewers will independently screen and extract data, with a third reviewer resolving 35
disagreements. 

3-)please be sure the keywords are precise.

4-)be sure that the references are directly related to the text.

5-)please mention if chatbots are utilized in the relevant section.

6-)be sure that the references are relevant:

However, the definition of chemsex remains multifaceted and subject to ongoing 69
debate in the scientific literature (Sousa; Camargo; Mendes, 2023; Chone et al., 2021, 70
Giorgetti et al., 2017) influenced by social, historical, and economic factors, as well as by 71
the availability of substances in different regions and time periods. I

7-)please add more references to support your claims.

 Nevertheless, chemsex is strongly associated with a range of biological 98
and psychosocial risks, including higher prevalence of sexually transmitted infections, 99
episodes of sexual violence, substance misuse and dependence, mental health disorders, 100
suicide attempts, and increased social vulnerability (Diehl et al., 2019; Hegazi et al., 2017

8-)be sure its true information. you may add reference to support

estricted set of substances (such as methamphetamine, GHB/GBL, mephedrone, and 143
ketamine), but this review aims to include any individual who associates the use of 144
psychoactive substances with sexual activity, irrespective of the substance used.

9-)be sure that ethical statement is appropriate.

10-)give more information about the review protocol

11-)explain novelty of your review protocol.

12-)please highlight novelty of your protocol in the abstract

13-) authors can also focus on the novelty of protocol in the abstract.

Author Response

. Protocol registration status
Response: We clarified in the Review Status section that the protocol has been submitted for publication and outlined the next steps (database searches, screening, etc.).

2. Sentence about reviewers
Comment: The sentence about independent reviewers can be shortened or moved.
Response: We revised the text, moving this description to Materials and Methods for conciseness.

3. Keywords
Response: Keywords were reviewed for precision; redundant terms were removed.

4. References
Response: We carefully cross-checked references to ensure all are directly related to the text. Additional references were added where needed to strengthen claims (e.g., risks associated with chemsex).

5. Chatbots
Response: A statement on the Use of Artificial Intelligence was added, clarifying that Gemini was used exclusively for language review, not content creation.

6–8. References and claims
Response: We reviewed the sections noted (definition of chemsex, associated risks, restricted set of substances) and ensured references are accurate and supported by peer-reviewed sources. Additional references were included to strengthen evidence.

9. Ethical statement
Response: We confirmed and clarified that, as per Brazilian Resolution 510/2016, scoping reviews do not require ethics approval.

10–13. Novelty of the protocol
Response: The novelty of this protocol was explicitly highlighted in both the Abstract and Introduction. We stressed that this is the first scoping review addressing healthcare access for chemsex in Brazil, mapping barriers, facilitators, and service gaps.

Reviewer 4 Report

Comments and Suggestions for Authors

Dear Authors,

Please, see the below few comments that you might consider or clarify:

You intentionally expand “chemsex” to include any substance used in sexual contexts (i.e., roll SDU into chemsex). This conflation will inflate heterogeneity and contaminate signals (e.g., alcohol-only SDU vs stimulant/GHB-centric chemsex).

Suggestion: Pre-specify operational definitions and strata: e.g., chemsex-narrow (meth/GBL/GHB/mephedrone/ketamine) vs SDU-broad (incl. alcohol, cannabis, poppers, etc.), with planned stratified extraction/analysis so findings aren’t over-generalized.

Inclusion is limited to 2014–2024, yet the references and access dates in the protocol itself are from 2025, and the review is being executed in 2025.

Suggestion: Extend to “inception–search date” or at least through 2025; pre-specify exact run dates and a search-update plan prior to submission.

The text states you will follow “PRISMA-ScR (Moher 2009)”—that’s the PRISMA (systematic review) paper, not PRISMA-ScR. You cite Tricco 2018 elsewhere, but the mixed messaging is an accuracy issue.

Fix: Align to Tricco et al., 2018 PRISMA-ScR throughout; keep Moher 2009 only if also using the original PRISMA diagram for transparency, but specify PRISMA-ScR as the reporting backbone.

You propose classifying “grade of recommendation/level of evidence (Oxford CEBM 2009).” Scoping reviews map evidence and typically do not grade strength or make recommendations. Using CEBM levels (treatment-focused, outdated version) risks over-interpreting diverse qualitative/observational literature and confusing readers.

suggestion: Either (a) drop grading entirely (preferred), or (b) justify a limited, descriptive appraisal using appropriate JBI tools by study design, clearly stating no inferential weighting/recommendations will be made. Do not translate to “grades of recommendation.”

JBI/Levac’s “consultation” refers to stakeholder engagement (e.g., clinicians, harm-reduction NGOs, community peers). You equate it to citation tracking only.

Suggestion: Reframe “consultation” as stakeholder consultation (co-design priorities, validate categories, identify grey literature), and keep citation tracking under “supplementary search methods.”

 Given the topic’s behavioral dimensions, I suggest Add PsycINFO at minimum.

Grey literature can be added?

 The introduction presents SUS as uniformly enabling nationwide access and being “essential” for harm-reduction delivery; in practice, availability and implementation vary widely by state/municipality and over time. Temper language and present this as a hypothesis to be mapped, not an assertion.

Statements that there is “little synthesized evidence on access” are plausible, but either support with a brief scoping of existing reviews (Brazil-specific) or phrase cautiously.

Chemsex vs SDU: As noted, avoid implying that findings from SDU (e.g., alcohol+sex) are interchangeable with stimulant/GHB-centric chemsex across all outcomes; plan strata to prevent over-generalization.

looking forward to reading your revised work

Best wishes

Author Response

1. Chemsex vs SDU definitions
Comment: Avoid conflating definitions. Suggest stratified approach.
Response: We accepted this suggestion. The manuscript now clearly distinguishes chemsex-narrow (stimulants, GHB/GBL, mephedrone, ketamine) and SDU-broad (alcohol, cannabis, poppers, etc.), with results to be stratified accordingly.

2. Time frame
Response: We extended the inclusion period through July 2025 and clarified that searches will be updated before final submission.

3. PRISMA-ScR citation
Response: We corrected inconsistencies: PRISMA-ScR is cited as Tricco et al., 2018 throughout, and Moher 2009 remains only for transparency regarding the diagram.

4. Evidence grading
Response: We removed the reference to CEBM grading, clarifying that as a scoping review, we will map evidence descriptively using JBI guidance without inferential weighting.

5. Consultation phase
Response: We reframed the Consultation section to emphasize stakeholder engagement (clinicians, NGOs, community peers) and relocated citation tracking to Supplementary Search Methods.

6. Databases and grey literature
Response: We added PsycINFO to broaden coverage.

7. Introduction—SUS
Response: We tempered statements on SUS, clarifying that while it is universal in principle, availability and implementation vary across regions and over time. This is now framed as a hypothesis to be mapped.

8. Evidence gaps
Response: We adjusted wording to note that “little synthesized evidence exists” and supported it with references to existing Brazil-specific studies.

We are grateful for the reviewers’ insightful comments. We believe the revisions made in response to each point have substantially improved the manuscript, increasing clarity, methodological rigor, and contribution to the literature.

Round 2

Reviewer 4 Report

Comments and Suggestions for Authors

Thank you for addressing the comments